# Do LLMs Really Understand Code? A Semantic Benchmark with Automated Question Generation and Evaluation

## Abstract

Large language models (LLMs) have demonstrated impressive results in code generation tasks, yet it is unclear to what extent they genuinely understand code semantics and whether this affects their ability to write high-quality code. To address this question, we introduce **SemBench**, a novel benchmark consisting of **1,000** diverse C programs sourced from the CodeParrot GitHub-code dataset, with **15,404** semantic questions spanning six fundamental properties: function reachability, loop reachability, data dependency, liveness of variables, dominator sets, and dead code. These six types of concepts are taught in undergraduate-level programming language classes and can be computed precisely and efficiently by deterministic algorithms. In contrast to existing benchmarks (e.g. HumanEval, MBPP, CodeXGLUE, SWE-bench) that emphasize code generation or functional correctness, our benchmark focuses on semantic understanding with deterministic answers. We evaluate **14** popular LLMs across **7** families—including GPT-4o Mini, GPT-3.5 Turbo, DeepSeek-Coder, CodeLlama, Qwen, StarCoder, Mistral, and Phi. To our surprise, they have very high failure rates, ranging from **21.40%** to **81.86%**. Category analysis reveals a sharp split between local control-flow and global data-flow reasoning and highlights performance divergence across task types, where different models excel on different categories. SemBench rankings demonstrate high correlation with HumanEval and MBPP, which proves its potential to be a good indicator of whether an LLM can produce high-quality code. In fact, further study shows that the LLMs under evaluation have difficulty even understanding their *own coding output*. For example, DeepSeek-Coder-V2-Lite-Instruct fails to identify variable liveness correctly 58.23% time. Overall, our experiments provide deeper insights into semantic understanding, reveal the substantial gap between semantics and code completion in modern LLMs, and open new opportunities for further improvements of coding LLMs.

## 1 Introduction

Large Language Models (LLMs) have achieved remarkable progress in code-related tasks, evolving significantly from demonstrating reasoning capabilities Chen et al. (2021), to instruction-tuned architectures Ouyang et al. (2022), and retrieval-augmented generation frameworks Lewis et al. (2020); Borgeaud et al. (2022). To systematically evaluate these advances, diverse benchmarks have emerged, predominantly assessing metrics like functional correctness, code generation accuracy, and semantic matching. For instance, HumanEval Chen et al. (2021) evaluates Python function generation with functional correctness, and CodeXGLUE Lu et al. (2021) aggregates several tasks across multiple programming languages. Such benchmarks have been instrumental in driving improvements in LLM performance. As LLMs progress toward becoming autonomous software development agents, one cares about the reliability and correctness of the code generated by LLMs. In addition to thorough testing (which is always limited in coverage), we ask the question "Do LLMs truly understand code semantics?"

Despite the flourishing ecosystem of code benchmarks, existing datasets often inadequately evaluate semantic understanding directly. Most benchmarks predominantly target syntax-level learning or the overall pass rates of generated programs, such as MBPP Austin et al. (2021) and APPS Hendrycks et al. (2021a). Although recent works have begun incorporating semantic evaluation, like SeqCoBench Maveli et al. (2025), they either rely heavily on manually collected labels or indirectly evaluate semantic understanding. Furthermore, semantic benchmarks continue to focus largely on high-level languages like Python, leaving languages like C comparatively neglected.

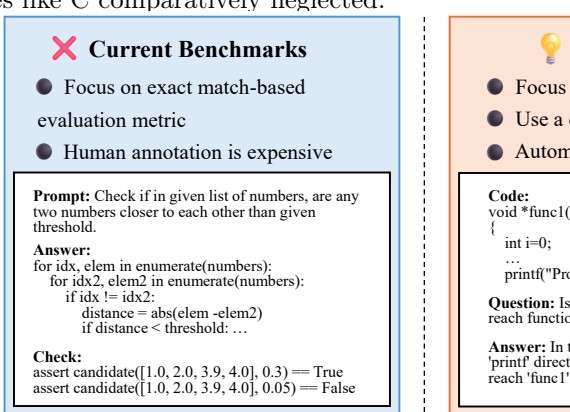

Figure 1: Comparison between current benchmarks and SemBench.

Evaluating semantic understanding at scale introduces unique challenges, particularly regarding question generation and output evaluation. Human-annotated labels are expensive and limit the scalability of training data Hendrycks et al. (2021a). Additionally, determining if an LLM truly "understands" code is intrinsically ambiguous, especially given the high uncertainty associated with LLM interactions Bubeck et al. (2023). To address these challenges, we propose the automated question generation-evaluation framework (SemBench-QE), reframing semantic evaluation as a binary question-answer task with explanations. As illustrated in Figure 1, our approach diverges significantly from popular benchmarks such as HumanEval. We begin by cleaning and filtering code samples from Hugging Face's Code-Parrot corpus Tunstall et al. (2022), and choosing semantically rich examples. Leveraging compiler frameworks like Clang's abstract syntax trees (ASTs) and LLVM Lattner and Adve (2004), or static program analysis, we derive 6 properties, including function reachability, loop reachability, data dependency, liveness of variables, dominator sets, and dead code, to evaluate the understanding of the data and control flow semantics, with automatic labeling for LLM evaluation. These properties are routinely taught in Computer Science programming or compiler courses, and are considered basic concepts in code understanding. These properties are instantiated into natural-language queries through predefined templates, yielding structured question–answer pairs with automatic ground truth; robust prompting and syntax-based parsers then validate model outputs, enabling scalable and reproducible benchmark construction. Using this pipeline, we present SemBench, comprising 15,404 crafted semantic questions about 6 semantic properties of 1000 C programs. To demonstrate generalizability, we also extend the pipeline and create a pivot set of 540 questions about 30 Python programs under the same 6 categories.

In summary, our contributions are threefold:

1. We introduce **SemBench**, the first large-scale benchmark that contains 15404 questions, which is explicitly designed to evaluate semantic code understanding, addressing a notable gap in prior research.

2. We introduce an extensible and fully automated question-generation and evaluation framework (SemBench-QE) that eliminates manual labeling and effectively handles uncertainties inherent in LLM outputs.

3. We conduct comprehensive empirical analyses connecting our benchmark to prior work on code reasoning and model evaluation, identifying connections between code understanding and code generation ability.

And our findings are summarized as follows:

1. **Deficiency in understanding.** many widely used models have around 40–70% error rate on SemBench, even the strongest model attains **21.40%**, leaving considerable headroom for improvement.

2. **Local vs. global semantics understanding.** Control-flow categories are near-solved for top models, but the global data-flow property *Liveness* presents a serious challenge with error rate **37.81% - 85.24%**, revealing a gap in *global* data-flow versus *local* control-flow understanding.

3. **Inconsistent reasoning patterns.** Model performance varies widely across categories: strengths in some semantic tasks are offset by weaknesses in others, underscoring the fragmented nature of current semantic understanding.

4. **Relation to code generation.** Aggregate ranks *positively correlated* with HumanEval/MBPP ($\rho$=0.61/0.72) in terms of Spearman's rho, indicating that SemBench evaluation result is a good indication of coding completion capability.

5. **Scaling is broadly effective, with rare exceptions.** Larger models generally achieve higher accuracy, confirming that scale enhances semantic reasoning. The code-specialized *StarCoder2* scales smoothly while the general-purpose *Qwen3* shows one exclusive anomaly given checkpoint instability and corpus imbalance.

6. **LLMs may not even fully understand the code they generate.** Overall accuracy on self-produced code does not improve over SemBench. This suggests that models may generate executable code without robust semantic comprehension.

## 2  PROBLEM SETUP

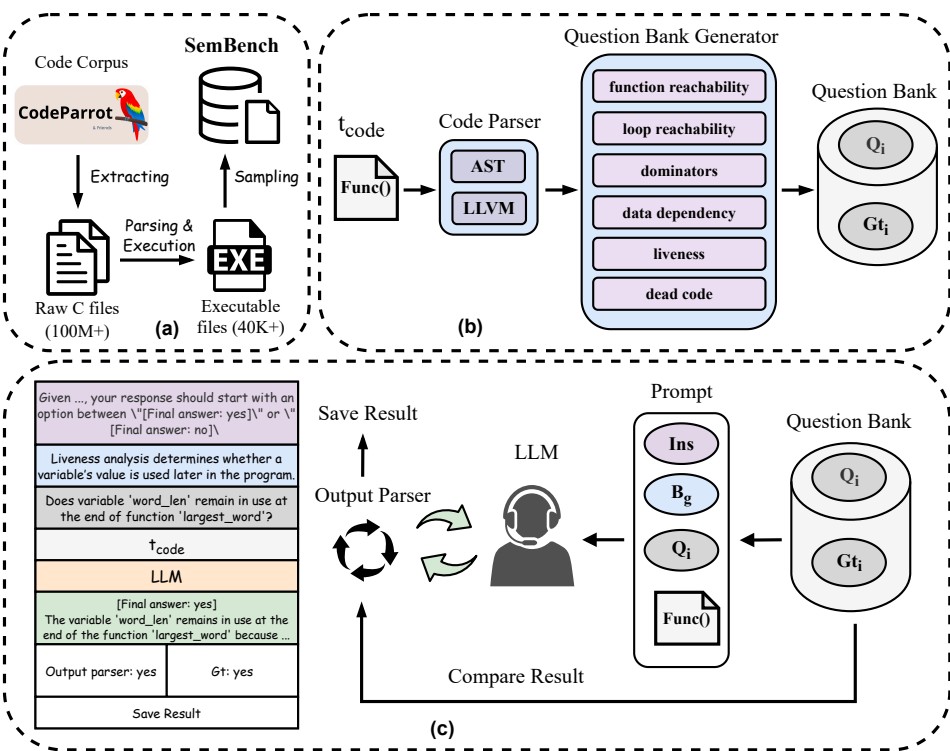

Figure 2: Overview of SemBench construction and SemBench-QE pipeline. Figure 2(a) describes how we collect, clean, and select C files to construct our raw code files; 2(b) represents the generation stage of the SemBench-QE pipeline; 2(c) describes the evaluation process with SemBench.

### 2.1  PROBLEM FORMULATION

Past works rely on manually collected labels, which restricts the size of the benchmark, while some others focus on LLM-generated responses, which are expensive to verify the correctness. To deal with these challenges, we aim to solve the following question:

Given the source code file $t_{code}$, how do we construct an automated pipeline to generate semantic questions and evaluate the LLM output?

Our automated question generation pipeline is shown in Figure 2(b). It first passes the cleaned $t_{code}$ to the code parser, which is built on top of AST and LLVM to generate accurate code semantic information. Then, the question bank generator, which produces six categories of semantic questions, takes in both of the semantic information and $t_{code}$ to produce pairs of questions $Q$ with real answer 'yes' or 'no', and ground truth label $G_t$ with value True/False. After constructing the question bank, we are able to evaluate LLM's understanding of these files' semantics using the generated $Q$–$G_t$ pairs.

Figure 2(c) demonstrates our evaluation process. The initial prompt $\mathbf{x}$ contains four components: instruction $Ins$, background information $B_g$, $t_{code}$ and question $Q$. The example prompts are listed in Appendix B. $Ins$ provides a general guidance that leads LLM to produce a response with the given format [Final answer: yes] or [Final answer: no]. And $B_g$ provides a brief but authoritative explanation of each category of questions. We then feed the initial prompt into the LLMs to get the response $O$.

$$\mathbf{x} = \mathrm{concat}\big(\langle \mathrm{BOS}\rangle,\ \tau(Ins),\ \tau(B_g),\ \tau(Q),\ \tau(t_{code})\big), \tag{1}$$

where $\tau(\cdot)$ tokenises a string into sub-word IDs and concat appends sequences left-to-right.

$$p_\theta(\mathbf{y} \mid \mathbf{x}) = \prod_{t=1}^{T} p_\theta\big(y_t \mid \mathbf{x},\ y_{<t}\big), \qquad y_t \in \mathcal{V}. \tag{2}$$

$$\hat{y}_t \sim p_\theta\big(\cdot \mid \mathbf{x},\ \hat{y}_{<t}\big), \quad t = 1, \ldots, T, \tag{3}$$

giving the final decoded sequence $O = \hat{\mathbf{y}} = \mathrm{LLM}_\theta\big(Ins, B_g, Q, t_{code}\big)$.

More details about the evaluation pipeline are included in Appendix A.4.

## 2.2 Research Questions

By designing such a pipeline, we aim to address the following research questions:

**Does an LLM truly understand code semantics?** Traditional benchmarks predominantly focus on code generation and completion tasks, such as HumanEval Chen et al. (2021) and MBPP Austin et al. (2021), however, they do not directly evaluate a model's semantic understanding. Recent benchmarks have started to shift focus towards semantics. For instance, PyX Ding et al. (2024) provides a dataset of executable Python programs with functional descriptions and execution traces, facilitating the training of models like SemCoder Ding et al. (2024) that reason about code semantics through monologue reasoning. Similarly, CodeMMLU Nguyen et al. (2024) offers a comprehensive multiple-choice benchmark assessing code comprehension across various programming languages. Nevertheless, these benchmarks either rely heavily on human-labeled data or do not directly evaluate semantic understanding. Our SemBench-QE framework enables the generation of question banks for any executable C files, allowing us to test LLMs' understanding with specific semantic patterns.

**How do different types of models perform in semantic understanding tasks?** Given the scarcity of benchmarks focusing on semantic understanding, most popular LLMs have not been extensively trained on semantic data. Meanwhile, various models have been introduced to the public, including general-purpose LLMs like ChatGPT-4o mini OpenAI (2024), reasoning-focused models like DeepSeek-R1 DeepSeek (2025), and a series of code-specific models. It is both interesting and important to explore their capabilities in code semantic tasks to understand their strengths and limitations.

**What is the relationship between code understanding and code generation abilities in LLMs?** While evaluating LLMs' semantic understanding is crucial, it is equally important to investigate the relationship between a model's understanding and its code generation capabilities. Can an LLM perform well in code generation tasks without truly understanding the underlying semantics?

## 3 Methods

```
1  int inc(int a){ return a+1; }
2  int sum_pos(int n){
3    int s = 0;
4    if (n <= 0) return 0;
5    // dominates early exit
6    for (int i=0;i<n;i++) {s+=inc(i);}
7    // may run if n>0
8    int k = 0;
9    while (k < 0) { s+=k; k++; }
10   // 0-trip => unreachable body
11   int t = s;
12   // def(t), no later use
13   if (0) { t = -1; } // dead branch
14   return s; // s live at exit
15 }
16 int main(){
17     int r=sum_pos(3);
18     printf("%d\n",r);
19 }
```

(a) Running example (trimmed).

(b) Illustrative graphs extracted.

Figure 3: Compact illustration of signals used by SemBench-QE: (a) code example; (b) extracted graphs.

### 3.1 Source File Collection and Preprocessing

To construct a benchmark that covers as much semantics as possible, and leverages the power of the compiler, we divided our collection and preprocessing into four steps. The first step is illustrated in Figure 2(a), to select representative source codes. We collect files from the CodeParrot GitHub Code dataset  Tunstall et al. (2022), which contains more than one million C files gathered from a large number of GitHub repositories. The statistics are listed in Appendix A.1. This guarantees that the base code files cover the wide semantics of real-life applications. The second step aims to select files that can be compiled and executed. To achieve this, we set two filters: contains the main function and does not import non-standard libraries, and obtain more than 150, 000 files post-selection. Afterwards, we verify the execution availability by executing these programs for four to five rounds and get more than 40, 000 files. Finally, to maintain the difficulty of the question bank, we selected 7792 programs by setting the complexity constraints e.g., loop count and computation count. 1000 files are finally selected, considering the resource limit.

### 3.2 Question Generation Pipeline

To solve the problem of time-consuming label collection and limited dataset capacity, our benchmark develops the SemBench-QE pipeline. It automates the whole process by systematically transforming $t_{code}$ into structured semantic information, accompanied by rigorously verified ground-truth answers. We rely on established static–analysis tools integrated within LLVM/Clang 17.0.6(LLVM Developers, 2024; Clang Team, 2024) to ensure the reliability and validity of the resulting dataset. Here, we provide a summary of the six categories, and full extraction details are deferred to Appendix §A.3.

- **Function Reachability (FR)** refers to whether function $f_1$ can (transitively) call $f_2$ in the static call graph. We collect ground truth by parsing with Clang AST to build a directed call graph and checking reachability via DFS/BFS. For example, in Fig. 3a, we could build the edges: main→sum_pos and sum_pos→inc, which yield the query "Is it impossible for sum_pos to reach inc?" with answer **No**.

- **Loop Reachability (LR)** captures whether a loop body executes at least once along some path. Ground truth is obtained from LLVM's SCEV trip-count reasoning, where zero iterations mark the loop body unreachable and otherwise it is considered reachable. In Fig. 3b, the loop `while(k<0)` is proven to have zero trips, leading to the query "Is the body of `while(k<0)` unreachable?" with answer **Yes**.

- **Dominance (Dom)** defines that a block $B$ dominates block $C$ if every path from function entry to $C$ passes through $B$. We derive ground truth by constructing the CFG and applying LLVM's Dominator Tree algorithm. As shown in Fig. 3b, the header `for(i<n)` dominates its body `s+=inc(i)`, producing the query "Does the loop header strictly dominate every block in its loop body?" with answer **Yes**.

- **Variable Data Dependency (Def–Use)** checks whether a variable has a use that occurs after its definition. We collect these relations by scanning Clang's AST tokens to pair definitions with later uses. For instance, Fig. 3a shows `s` defined before the loop and used both inside the loop and at the return statement, leading to the query "Is `s` used after its definition in `sum_pos`?" with answer **Yes**.

- **Variable Liveness** indicates whether a variable still holds a meaningful value at the end of a function. We approximate this through conservative backward data-flow analysis around function exit points. In Fig. 3b, `s` is live since it is returned, while `t` is not, giving rise to the query "Is `t` live at the end of `sum_pos`?" with answer **No**.

- **Dead Code (DC)** refers to statements that are unreachable under any execution path. We collect labels using Clang's path-sensitive Static Analyzer. In Fig. 3b, the branch `if(0){t=-1;}` is dead, yielding the query "Are the statements inside `if(0){}` dead?" with answer **Yes**.

## 3.3 Extension to Runtime-Interpreted Coding Languages

While the original SemBench-QE for C relies on Clang/LLVM to extract AST, CFG, SSA, and dataflow properties, we extend the framework to Python by substituting those compiler-based analysis tools with interpreter-side analysis tools. In place of the Clang AST, we use Python's `ast` (or LibCST); for CFG and SSA, we employ frameworks such as Scalpel, and for interprocedural reachability, we adopt call-graph analyzers like PyCG. These substitutions allow us to recover the same semantic categories. Importantly, static program analyses are independent of whether a language is compiled or interpreted, as they are defined over program structure rather than execution model (Nielson et al., 1999; Alfred V. Aho et al., 2007).

## 4 Experiments

Table 1: Statistics of LLMs and their Pass@1 (%) on HumanEval and MBPP. "–" denotes no official score published.

| Family | Model Name | Type | Size (B) | HumanEval | MBPP |
|---|---|---|---|---|---|
| GPT | GPT-4o Mini | General | – | 88.4 | – |
| | GPT-3.5 Turbo | General | – | 71.9 | – |
| DeepSeek | DeepSeek-Coder V2-Lite-Instr | Code | 16 | 81.1 | 68.8 |
| | DeepSeek-Coder 7B-Instr v1.5 | Code | 7 | 64.1 | 64.6 |
| | DeepSeek-R1-Distill-Qwen-7B | Reasoning | 7 | – | 17.2 |
| Llama | CodeLlama-13B-Instr | Code | 13 | 42.7 | 49.4 |
| | CodeLlama-7B-Instr | Code | 7 | 34.8 | 44.4 |
| | Llama-3 8B-Instr | General | 8 | 72.6 | – |
| Mistral | Mistral 7B-Instr (v0.3) | General | 7 | – | – |
| | Mamba Codestral 7B (v0.1) | Code | 7 | 75.0 | 68.5 |
| Qwen | Qwen2.5-Coder 14B-Instr | Code | 14 | 89.6 | 86.2 |
| | Qwen3 14B | General | 14 | – | 73.4 |
| StarCoder | StarCoder 2 7B | Code | 7 | 30.5 | 47.4 |
| Phi | Phi-4 Reasoning (14B) | Reasoning | 14 | – | 12.5 |

## 4.1 EXPERIMENT SETUP

**Model selection** We evaluate SemBench on 14 popular state-of-the-art LLMs across 7 different families to cover a wide range of architectures. We lean to LLMs trained on code, but also cover reasoning and general models for comparison. Given the Q&A nature of SemBench, we also include many instructed versions.

**Experiment Setting** We evaluated all open-source language models with the HuggingFace Transformers text-generation API (v 4.41) and queried proprietary GPT models via the OpenAI Python SDK (openai v 1.15). Experiments ran on a SLURM-managed node equipped with four AMD Instinct MI210 GPUs under the ROCm 5.7 stack. Exact hyperparameters and launch commands are provided in the scripts.

## 4.2 KEY FINDINGS

**Overall Performance** As shown in Table 2, selected models perform varies across SemBench. GPT family outperforms other models, demonstrating its competitive ability across diverse tasks. While a larger model size usually indicates an advantage, Phi-4 Reasoning performs even worse than 7B models. This may relate to model types and whether being instructed. Although code models may not be trained over semantic benchmarks, they usually achieve better performance. On the other hand, given the question format, instructed models tend to have a better performance. Overall, it is alarming to see that some of the properties, such as liveness, are poorly understood by all models under evaluation, with less than 50% correctness for most of them (note that random guesses can achieve 50% accuracy on average).

Table 2: Overall and per-category accuracy (%) on SemBench.

| Family | Model | Overall Accuracy | Data-Dep | Dead Code | Dominators | Func-Reach | Liveness | Loop-Reach |
|--------|-------|-----------------|----------|-----------|------------|------------|----------|------------|
| GPT | GPT-4o Mini | 78.60 | 82.97 | 99.35 | 100.00 | 87.99 | 36.79 | 67.56 |
| | GPT-3.5 Turbo | 71.99 | 66.13 | 71.95 | 95.22 | 67.59 | 37.71 | 84.50 |
| DeepSeek | DS-Coder V2-Lite | 68.71 | 78.81 | 81.70 | 47.63 | 77.79 | 50.99 | 68.17 |
| | DS-Coder 7B v1.5 | 48.09 | 41.98 | 76.45 | 45.14 | 41.18 | 36.23 | 50.41 |
| | DS-R1-Distill | 17.22 | 34.87 | 9.05 | 4.27 | 14.40 | 22.14 | 18.98 |
| Llama | CodeLlama-13B-Instr | 42.65 | 46.69 | 56.45 | 31.50 | 45.38 | 39.64 | 38.19 |
| | CodeLlama-7B-Instr | 47.99 | 80.31 | 30.70 | 12.06 | 52.31 | 62.19 | 47.35 |
| | Llama-3 8B-Instr | 40.35 | 37.42 | 53.75 | 23.51 | 48.56 | 35.47 | 38.45 |
| Mistral | Mistral 7B-Instr | 65.50 | 67.79 | 68.70 | 59.69 | 70.12 | 52.21 | 68.07 |
| | Codestral 7B v0.1 | 44.09 | 37.02 | 62.50 | 38.98 | 54.82 | 33.38 | 36.54 |
| Qwen | Qwen2.5 14B Instr | 53.09 | 50.65 | 81.90 | 73.33 | 49.38 | 25.34 | 46.77 |
| | Qwen3 14B | 42.94 | 27.61 | 83.55 | 48.50 | 58.40 | 19.85 | 24.89 |
| StarCoder | StarCoder 2 7B | 33.60 | 39.98 | 37.65 | 17.20 | 36.08 | 37.15 | 32.49 |
| Phi | Phi-4 Reasoning | 18.14 | 21.24 | 17.20 | 8.60 | 23.73 | 14.76 | 18.45 |

**Performance Over Different Categories** Table 2 reveals a sharp contrast between "shallow" and "deep" semantic tasks.

*Dead Code* and *Dominators* are almost trivial for GPT-4o (99–100% *accuracy*) and remain above 70% even for 7 B-parameter coders. These patterns can be solved by local syntactic cues, suggesting that giant language models already act as powerful static analysers for intra-procedure control flow. *Liveness* is the clear bottleneck: GPT-4o drops to **36.79%** *accuracy*, and every other model also falls. Correct reasoning requires tracking variable definitions across branches and loops, a form of classic data-flow analysis that remains elusive for purely autoregressive models. *Func-Reach* scores straddle the middle ground (88% vs. 68% for GPT-4o and Mistral-7B-Instr, respectively), indicating partial but incomplete mastery of call-graph reasoning.

Taken together, the plateau on "easy" tasks but steep drop on "deep" tasks implies that current LLMs internalise many surface semantic signals yet do *not* possess a unified, compiler-level understanding of program meaning. SemBench therefore fills a critical gap by exposing exactly where that boundary lies. We also use the extension SemBench to generate a Python pilot; the result is in Appendix C.3, which aligns with the C version we present here.

**Ablation Study Over Benchmark** To further validate the effectiveness of *SemBench*, we conducted an intra-family scaling study over both the Qwen3 (general-purpose) and

StarCoder2 (code-specialized) model families. As shown in Figure C.1, the overall accuracy improves with model size increases, confirming the expected scaling-law effect.

Interestingly, the scaling behaviors differ. For Qwen3, the 4B model underperforms the smaller 1.7B variant (16.6% vs. 20.0%), likely due to optimization instability or category imbalance, as accuracy on *Dominators* drops sharply at 4B. In contrast, StarCoder2 scales more smoothly, reflecting its code-focused training pipeline and better alignment with the program-analysis tasks in *SemBench*.

### 4.3 Relationship between Code Semantic and Generation

Table 3: Spearman correlation between SemBench and two code–generation benchmarks.

| Category | HumanEval | | MBPP | | |
|---|---|---|---|---|---|
| | $\rho$ | $p$ | $\rho$ | $p$ | $N$ |
| All Categories | 0.61 | 0.060 | 0.73 | 0.016* | 10 |
| DataDep | 0.21 | 0.556 | 0.28 | 0.425 | 10 |
| DeadCode | 0.83 | 0.003* | 0.96 | 0.000* | 10 |
| Dominators | 0.75 | 0.013* | 0.98 | 0.000* | 10 |
| FuncReach | 0.64 | 0.048* | 0.75 | 0.013* | 10 |
| Liveness | −0.52 | 0.128 | 0.04 | 0.907 | 10 |
| LoopReach | 0.36 | 0.310 | 0.47 | 0.174 | 10 |

\* indicates significant correlation at $p < 0.05$.

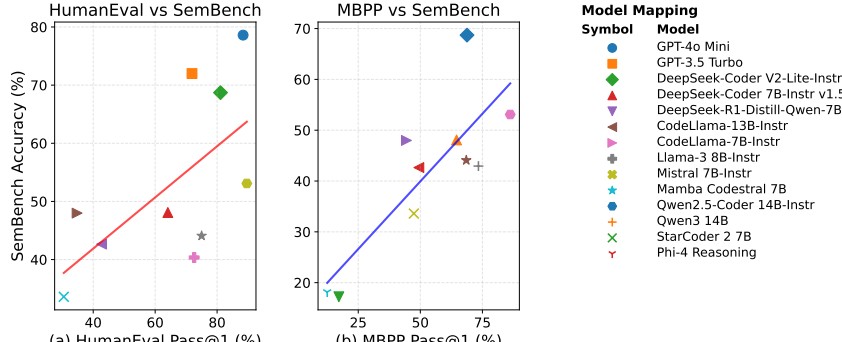

Figure 4: Scatter Plot of SemBench vs HumanEval and SemBench vs MBPP

**Correlation Analysis.** To examine whether *SemBench* performance transfers to established code–generation benchmarks, we compute rank–based correlations between each *SemBench* accuracy and the official *pass@1* reported for **HumanEval** and **MBPP**. Let $s_i^{(k)}$ denote the *accuracy* of model $i$ on *SemBench* category $k$ (including the overall SemBench), and $h_i$, $m_i$ be its HumanEval and MBPP pass@1 scores, respectively. We report

$$\rho_k^{\text{HE}} = \text{Spearman}\big(\{s_i^{(k)}\}, \{h_i\}\big), \qquad \rho_k^{\text{MBPP}} = \text{Spearman}\big(\{s_i^{(k)}\}, \{m_i\}\big),$$

together with Kendall's $\tau$ and two–sided $p$–values. Spearman's $\rho$ is widely adopted for cross-benchmark studies because it is robust to non-linear scaling and outliers (Liang et al., 2022), while Kendall's $\tau$ mitigates small-sample bias (Zan et al., 2025). Models lacking an official score for either benchmark are filtered out pair-wise.

**Overall Trends Demonstrate Correlation Between SemBench and Code Generation Tasks.** Table 3 demonstrates a *moderate–to–strong* association: $\rho = 0.61$ with HumanEval ($p = 0.060$) and $\rho = 0.72$ with MBPP ($p = 0.019$). Thus, Figure 4 reveals that models that rank highly on SemBench tend to do well on established code-completion tasks.

**Local control-flow categories align strongly.** Dead Code and Dominators correlate tightly with both HumanEval ($\rho = 0.83$ and $0.75$) and MBPP ($\rho = 0.96$ and $0.98$, all $p < 0.01$), reflecting the local control-flow reasoning is included in the code-completion benchmarks.

**Global data-flow categories expose poor code understanding.** Data Dep and Loop Reach reveal significant weakness not covered in the existing coding benchmarks ($|\rho| \leq 0.29$, $p > 0.17$), while Liveness even correlates *negatively* with HumanEval ($\rho = -0.52$) and is essentially uncorrelated with MBPP ($\rho = 0.04$). These tasks, therefore, underscore the lack of semantic understanding not covered in the existing coding examples.

**Can LLMs understand the semantics of the code they generate?** To address this problem, we selected 120 problems from the **CodeContest** dataset (Li et al., 2022), prompted **DeepSeek-Coder V2-Lite-Instruct** to generate solutions, and retained 40 programs that executed successfully. We then evaluated these model-generated programs using our pipeline SemBench-QE.

DeepSeek-Coder does not gain accuracy over semantic understanding when tested on its own outputs. The model achieves an overall accuracy of **68.3%** over 612 semantic questions, similar to its performance on SemBench (Table 1). While relatively shallow categories such as *Dead Code* and *Dominators* are nearly solved, the accuracy even decreases for deeper semantic properties, including *Liveness* (**41.8%**) and *Loop Reachability* (**54.9%**). This highlights that although the model can generate executable code, it may struggle to reason about the long-range semantic properties of that same code.

## 5 Related Work

### 5.1 LLM and Code Benchmarks

Current benchmarks for code-related LLM evaluation largely fall into two categories: code generation and debugging. Datasets like HumanEval Chen et al. (2021) and MBPP Austin et al. (2021) evaluate Python function synthesis via docstrings and test cases, while CodeXGLUE Lu et al. (2021) expands to multiple languages and tasks such as clone detection and code repair. These benchmarks prioritize syntactic correctness and execution-based metrics but do not directly probe semantic understanding. A smaller subset addresses semantics indirectly through classification tasks like vulnerability detection or API misuse, relying on high-level labels rather than fine-grained reasoning about behaviors like data flow or reachability. Moreover, most focus on Python, with limited support for low-level languages like C. To our knowledge, no public benchmark directly tests LLMs using closed-form semantic questions, motivating our C-based benchmark designed to evaluate fundamental program understanding.

### 5.2 Code-oriented Models

Recent advancements in LLMs have led to a variety of code-focused models, each trained on distinct corpora and optimized for different tasks. **Code Llama** (7–13B) Rozière et al. (2024) is fine-tuned from LLaMA using a code-heavy corpus (85% source code) and includes instruction-tuned variants with synthetic Q&A pairs. **DeepSeek-Coder** DeepSeek-AI et al. (2024) includes dense (v1.x) and MoE (V2-Lite) models, trained on up to 6T tokens across 338 languages, with instruction tuning for efficient code QA. **StarCoder 2** Lozhkov et al. (2024) (3–15B), built on The Stack v2, shows strong performance on C reasoning, with the 15B model rivaling larger baselines. **Qwen 2.5-Coder** Hui et al. (2024) adapts a multilingual base model using a code-centric corpus across 92 languages, with the 14B instruct variant offering extended context and strong tool-calling support. **Codestral-22B** combines instruction tuning with 80+ languages Jiang et al. (2023).

## 6 Conclusion

We presented SEMBENCH, a benchmark for directly evaluating LLMs' semantic understanding through automated question generation and static features-based ground truths. Our study shows that while local control-flow properties (e.g., dominators, dead code) are nearly solved, deeper global data-flow tasks (e.g., liveness) remain difficult, exposing substantial room for improvement. Model scaling generally helps but is inconsistent across families, and evaluation on self-generated code reveals that producing runnable programs does not imply semantic comprehension. SemBench further correlates with HumanEval and MBPP, highlighting semantic reasoning as a key factor in code generation quality. We hope it serves as both a diagnostic and a resource for developing models with stronger program semantics.

*The anonymous link to SemBench is released in the supplementary material.*

**Reproducibility.** We make every effort to ensure reproducibility: dataset construction (Sec. 3), question-generation pipeline (Sec. A.3), and evaluation procedure (Sec. A.4) are fully detailed in the main paper and Appendix, with implementation scripts provided in the supplementary material. All code and processed data will be released under an open license.

**Ethics Statement.** This work does not involve human subjects, personal data, or sensitive attributes. All datasets are derived from publicly available GitHub code with license compliance. We acknowledge potential dual-use concerns, as semantic benchmarks may expose weaknesses in models; however, we intend to advance safer and more reliable LLMs for code understanding.

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

## A  Benchmark

### A.1  SemBench Statistics

Table A.1 provides an overview of key attributes of SemBench, including the number of programs, the total number of questions (problems), and the count of questions in each semantic category.

Table A.1: Overview of the SemBench

| Question Bank Size | | |
| --- | --- | --- |
| Program counts | | 1 000 |
| Problem counts | | 15 404 |
| **Problem counts by category** | | |
| Data dependency | | 1 996 |
| Dead code | | 2 000 |
| Dominators | | 1 965 |
| Function reachability | | 3 548 |
| Liveness | | 1 965 |
| Loop reachability | | 3 930 |
| **Feature statistics** | | |
| File size (bytes) | 3 108 | (min: 85, max: 491 196) |
| Max nesting depth | 2 | (min: 0, max: 10) |
| Avg dependency distance | 21 | (min: 0, max: 630) |
| Max dependency distance | 67 | (min: 0, max: 2 071) |

In Table A.1, each category lists the number of Yes/No questions (problems) that test that aspect of code semantics.

### A.2  Notations in Tables

For brevity, we use shorthand labels for model names in the tables of the main paper. Table A.2 provides a mapping between these shorthand labels and the models' full repository or API identifiers.

Table A.2: Shorthand mapping in other tables and full model identifiers.

| Shorthand in other tables | Full repo/API identifier |
| --- | --- |
| GPT-4o Mini | `openai/gpt-4o-mini` (chat.completions API) |
| GPT-3.5 Turbo | `openai/gpt-3.5-turbo` (chat.completions API) |
| DeepSeek-Coder V2-Lite-Instr | `deepseek-ai/DeepSeek-Coder-V2-Lite-Instruct` |
| DeepSeek-Coder 7B-Instr v1.5 | `deepseek-ai/deepseek-coder-7b-instruct-v1.5` |
| DeepSeek-R1-Distill-Qwen-7B | `deepseek-ai/DeepSeek-R1-Distill-Qwen-7B` |
| CodeLlama-13B-Instr | `codellama/CodeLlama-13b-Instruct-hf` |
| CodeLlama-7B-Instr | `codellama/CodeLlama-7b-Instruct-hf` |
| Llama-3 8B-Instr | `meta-llama/Llama-3-8B-Instruct` |
| Mistral 7B-Instr (v0.3) | `mistralai/Mistral-7B-Instruct-v0.3` |
| Mamba Codestral 7B (v0.1) | `mistralai/Mamba-Codestral-7B-v0.1` |
| Qwen2.5-Coder 14B-Instr | `Qwen/Qwen2.5-Coder-14B-Instruct` |
| Qwen3 14B | `Qwen/Qwen3-14B` |
| StarCoder 2 7B | `bigcode/starcoder2-7b` |
| Phi-4 Reasoning (14B) | `microsoft/Phi-4-reasoning` |

### A.3  Ground Truth Collection Methods

**Function Reachability** Function reachability queries explore whether a function can potentially invoke another within a call graph derived from static analysis. The pipeline is able to utilize tools such as Clang's Abstract Syntax Tree (AST) parsing capabilities(LLVM Project, 2024a) to accurately identify function declarations and calls. A directed call graph is constructed where nodes represent functions and edges denote explicit call relationships.

Reachability checks (e.g., depth–first or breadth–first search) provide precise ground–truth answers, as C's static call semantics exclude indirect calls unless explicitly encoded via function pointers(Alfred V. Aho et al., 2007). Consequently, ground–truth answers for function reachability are exact, reflecting definitive interprocedural call paths. As shown in Figure 1, the question template is "Is it impossible for function $fun1$ to reach function $fun2$?", The call graph connectivity provides us with ground-truth, and we save the $fun1$, $fun2$, and $G_t$ as the semantic information to fill into the template.

**Loop Reachability** Loop reachability determines whether loops can potentially be skipped or are guaranteed to be executed. We employ LLVM's Scalar Evolution (SCEV) analysis(LLVM Project, 2024c) to identify loops with determinable trip counts. If the maximum iteration count computed by SCEV is zero, loops are marked unreachable; otherwise, loops are conservatively considered reachable. The correctness of Scalar–Evolution analysis—widely used in compiler optimisations—is well established(Steven S. Muchnick, 1997), rendering our ground-truth labels robust.

**Dominance Analysis** At the code level, dominance captures mandatory execution order between program blocks. A block $B$ is said to dominate block $C$ if every execution of $C$ must first pass through $B$. In practice, this means that (i) function entries dominate all statements in the function, (ii) loop headers dominate their loop bodies, and (iii) branch conditions (e.g., `if`) dominate the statements within their guarded regions. LLVM's Dominator Tree analysis(LLVM Project, 2024b), based on the Lengauer–Tarjan algorithm(Lengauer and Tarjan, 1979), computes these exact relationships, and we instantiate them into binary queries such as "does the loop header dominate every block within its loop body?".

**Variable Data Dependency** Data dependency queries examine whether variables are used after their definitions, capturing essential def–use relationships within the program. The pipeline leverages Clang's AST–based token analysis to accurately identify variable definitions and subsequent uses(Steven S. Muchnick, 1997). Variables identified as having a use after definition are explicitly marked. The correctness and completeness of libclang's AST parsing ensure accurate representation of data dependencies, making these labels trustworthy. One example is shown in Figure 2(c), given the extracted $var$ and $fun$ pair, this generator can generate a rich combination of data dependency questions.

**Variable Liveness** Variable liveness analysis identifies variables still holding meaningful values ("live") at the exit points of functions. Our pipeline approximates standard data–flow analysis(Keith D. Cooper and Linda Torczon, 2012) by examining explicit variable usage near function–exit points. A variable is conservatively considered live if it appears after its final definition within the function's terminating region. This heuristic approach, though conservative, closely aligns with conventional compiler–liveness analysis(Steven S. Muchnick, 1997), ensuring reliable ground truth.

**Dead Code Detection** Dead code detection queries determine the executability of program statements. Our pipeline employs Clang's Static Analyzer, specifically its interprocedural, path–sensitive dead–code checker(Clang Static Analyzer Team, 2024). Statements flagged by the analyzer as unreachable under any execution path are marked as dead. The Static Analyzer's rigorous and extensively validated techniques(Ted Kremenek et al., 2008) guarantee high reliability for these ground–truth labels.

**Task Design Mindmap** To make SemBench a robust and decomposable benchmark, we intentionally assign different difficulty levels to different categories of tasks. Dead code and Dominators tasks are relatively easy, as they do not need too much information. Liveness and Data Dependency are relatively hard as they may rely on understanding the whole program. Users are free to explore the combinational and decompositional application of SemBench.

On the other hand, even for codes that have been exposed to LLMs, these semantic features would not be leaked. Experiment in Section 4.3 proves that LLMs could fail to improve understanding even for the programs generated by themselves.

## A.4 Evaluation Pipeline

As shown in Figure 2(c), we also built the automated evaluation pipeline. Same as discussed in section 2.1, we concatenate four components to form the rich-context initial input from the constructed question bank. Then we feed the initial input into the LLM to get $O$. Afterwards, we will pass $O$ through our well-designed output parser to extract the LLM's answer.

The output parser is designed to reduce mislabeling outputs produced by LLMs.

**Output Parser**

To guarantee the correctness of the evaluation, parsing $O$ correctly matters. Our output parser processes $O$ by following these steps:

- Cleaning non-English characters and non-instructed symbols.
- Search for and collect declared patterns in the $Ins$ within each line.
- Remove ambiguous answers, e.g., yes or no when it does not clearly answer.

Besides syntactically verifying its correctness, we also tested the parser over sixty real outputs generated by Phi-4-reasoning, which tends to produce diverse, noisy outputs. And our output parser passes all the tests.

Overall, our automated analysis pipeline combines rigorously validated compiler analyses(LLVM Developers, 2024; Clang Team, 2024; LLVM Project, 2024c;b; Clang Static Analyzer Team, 2024) and established compiler theory(Alfred V. Aho et al., 2007; Steven S. Muchnick, 1997; Keith D. Cooper and Linda Torczon, 2012) to produce a semantically accurate, robust benchmark dataset suitable for evaluating code–semantic understanding by Large Language Models (LLMs).

# B Prompt Engineering

## B.1 Full Prompts

As described in Section 2.1, each prompt to the LLM is constructed by concatenating four parts:

$$\big(Ins,\ B_g,\ t_{\text{code}},\ Q\big),$$

where

- $Ins$ is the *instruction* to the model to format the output;
- $B_g$ is the *background information* describing the category of each question;
- $t_{\text{code}}$ is the source C code;
- $Q$ is the specific question text (generated via the templates from Section 3.2).

Below we give the full text of the $Ins$ and $B_g$.

**Instruction ($Ins$):**

> Given the C code and background information, your response should start
> with either: `[Final answer: yes]` or `[Final answer: no]`, and then
> briefly explain your reasoning step by step.

**Background information ($B_g$) for each semantic category:** We supply the model with relevant definitions for each task. The background prompts are as follows:

function reachability   Examines whether one function can transitively call another (i.e., if a call path exists).

loop reachability   Examines whether a loop is executed during the program execution based on its condition and structure.

dominators   Identifies whether certain code sections (like loop headers) must execute before other parts of the loop.

| data dependency | Determines if a variable's defined value is used and influences subsequent computations. |
| --- | --- |
| liveness | Determines whether a variable's value is used later in the program. |
| dead code | Identifies segments that are never executed, such as code following an unconditional return or unreachable branches. |

## B.2 Ablation Study of Prompt and Explanation

While SemBench relies on the automated framework to collect judgments made by models, prompts to generate responses, and explanations matter in the response quality.

We conducted experiments to determine the effect of prompt style and token limits. A naive $\mathbf{Ins}_{\text{naive}}$ prompt simply asks for decision:

> Given the C code and background information, answer each test question
> with an option between [yes] and [no].

However, we found that this often produces invalid answers (e.g. missing the required tags). Therefore, we experimented with a Chain of Thought (CoT) version $\mathbf{Ins}_{\text{post}}$:

> Given the C code and background information, your response should start
> with an option between `[Final answer: yes]` or `[Final answer: no]`,
> then you should explain your solution step by step briefly.

Shifting from $\mathbf{Ins}_{\text{naive}}$ to $\mathbf{Ins}_{\text{post}}$ with a large token limit substantially increases performance: for example, increasing Phi4-reasoning from 12.54% to 18.14% and CodeLlama-7b-Instruct-hf from 37.30% to 47.99%.

While the $\mathbf{Ins}_{\text{post}}$ prompt enhances overall performance, the maximum token limit of 256 tokens results in wasted resources, as many outputs conclude their reasoning early but continue generating unnecessary tokens. Moreover, evaluating each 7B model fully requires over 8 hours. To address these issues, two solutions are promising: reducing the maximum token length or modifying the prompt to let LLMs answer in the beginning. Reducing the token limit can prevent excessive token generation, but risks truncating valid answers. Alternatively, switching to the answer-first style $\mathbf{Ins}$ prompt from Section B.1 provides greater flexibility regarding token limits.

To balance efficiency and performance, we conducted an ablation study on batch size and prompt effects. As shown in Table B.1, prompting LLMs to answer only after completing their reasoning causes a sharp performance drop when reducing the maximum token count from 128 to 64. Conversely, adopting the $Ins$ prompt preserves good performance while significantly improving processing speed.

Table B.1: Overall Accuracy (%) on SemBench for different batch-size / prompt settings

|  | $Ins_{post}\mathbf{64}$ | $Ins_{post}\mathbf{128}$ | $Ins_{post}\mathbf{256}$ | $Ins\mathbf{64}$ |
| --- | --- | --- | --- | --- |
| SemBench | 11.84 | 42.92 | 44.70 | 44.72 |

Therefore, SemBench is by no means a shallow Yes/No task, but provides tangible insights into LLMs' performance over code semantics understanding.

## C Experiments

### C.1 Intra-family Scaling Result

Figure C.1 shows the overall accuracy and per-category accuracy of Qwen3 and StarCoder2 families.

The result of this experiment, along with the nature of the semantic tasks we have constructed, proves the robustness and data-contamination-free nature of SemBench.

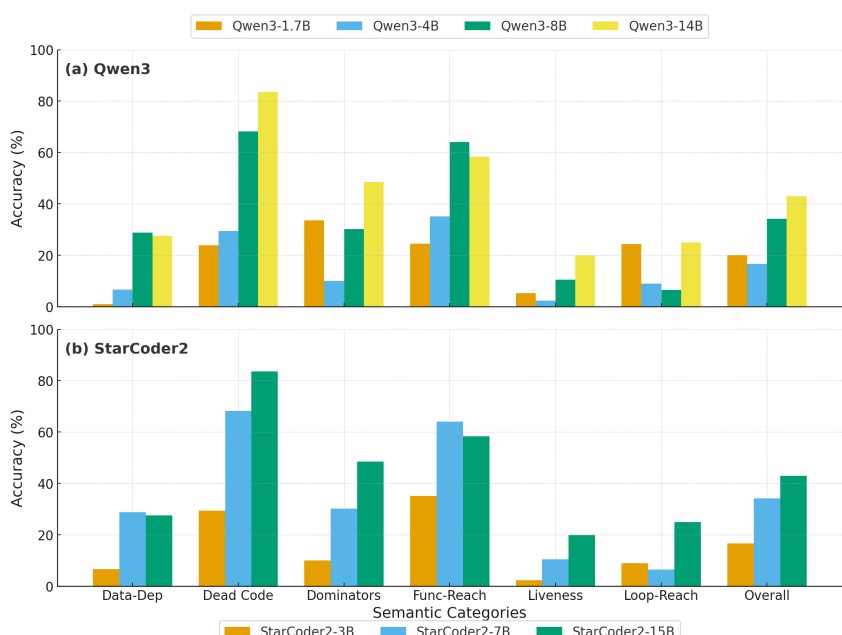

Figure C.1: Qwen3 and StarCoder2 family accuracy scaled by model size

Table C.1: Correlation between SemBench categories and two code–generation benchmarks.

| | HumanEval | | | | MBPP | | | | |
|---|---|---|---|---|---|---|---|---|---|
| **Category** | $\rho$ | $p$ | $\tau$ | $p$ | $\rho$ | $p$ | $\tau$ | $p$ | $N$ |
| SemBench | 0.61 | 0.060 | 0.47 | 0.073 | 0.73 | 0.016 | 0.60 | 0.017 | 10 |
| DataDep | 0.21 | 0.556 | 0.11 | 0.727 | 0.28 | 0.425 | 0.20 | 0.484 | 10 |
| DeadCode | 0.83 | 0.003 | 0.64 | 0.009 | 0.96 | 0.000 | 0.87 | 0.000 | 10 |
| Dominators | 0.75 | 0.013 | 0.60 | 0.017 | 0.98 | 0.000 | 0.91 | 0.000 | 10 |
| FuncReach | 0.64 | 0.048 | 0.51 | 0.047 | 0.75 | 0.013 | 0.60 | 0.017 | 10 |
| Liveness | -0.52 | 0.128 | -0.42 | 0.108 | 0.04 | 0.907 | -0.02 | 1.000 | 10 |
| LoopReach | 0.36 | 0.310 | 0.24 | 0.381 | 0.47 | 0.174 | 0.38 | 0.156 | 10 |

## C.2  COMPLETE CORRELATION STUDY

In Section 4.3, we reported overall rank correlations between SemBench and the code-generation benchmarks HumanEval and MBPP. Table C.1 gives the full Spearman ($\rho$) and Kendall ($\tau$) correlations for each SemBench category. Figure C.2 visualizes these Spearman correlations for all categories. To demonstrate the effectiveness, Figure C.3 plots the relationship between Dead Code accuracy and HumanEval pass@1, illustrating their strong association.

Based on these per-category results, we define an adjusted SemBench metric (Adj. `accuracy`) by retaining only the highly correlated categories. Specifically, we include Dead Code, Dominators, and Function Reachability (each having Spearman $\rho > 0.5$ with the benchmarks), and exclude categories with low correlation (e.g., Liveness, Loop Reachability). This adjusted metric focuses on the semantic tasks most aligned with code-generation performance.

Table C.3 presents the comparison between Adj. Accuracy with code generation benchmarks. Keeping categories with high correlation, the ranking of SemBench tends to be more similar to HumanEval and MBPP. Table C.4 shows that the Adj. SemBench has a higher correlation with code generation benchmarks. All of this shows that SemBench has the flexibility to adjust to not only overall code understanding tasks but also code generation tasks.

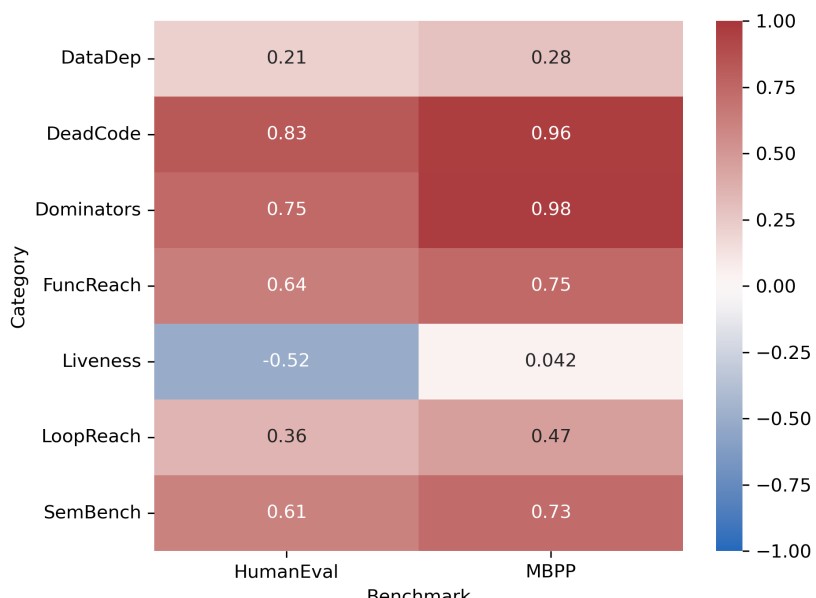

Figure C.2: Spearman $\rho$ between SemBench@1 and HumanEval / MBPP

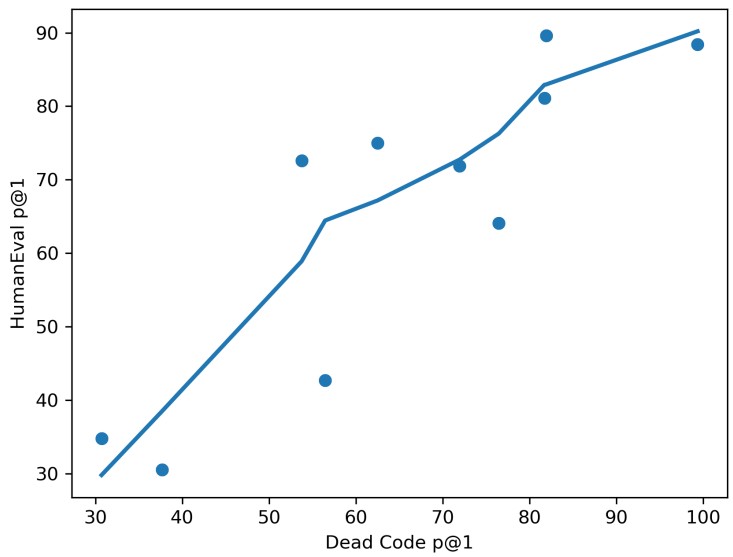

Figure C.3: Relationship between dead code problems and HumanEval

## C.3  SEMBENCH PYTHON EXTENSION RESULT

To show generality, we create a Python version pilot SemBench and run experiments on it. The results are shown in Table C.2. Here are some key findings

1. Overall accuracies over Python are higher than C.

2. The models that perform well over C keep dominating performance over Python.

Table C.2: Model accuracy and ranking on SemBench-C and SemBench-Python.

| Model | C accuracy | Py accuracy | C rank | Py rank |
|---|---|---|---|---|
| DeepSeek V2-Lite-Instr | 68.71 | 81.85 | 1 | 3 |
| Mistral 7B-Instr v0.3 | 65.50 | 83.52 | 2 | 1 |
| Qwen2.5-Coder 14B-Instr | 53.09 | 82.04 | 3 | 2 |
| CodeLlama-7B-Instr | 47.99 | 67.04 | 4 | 6 |
| Qwen3 14B | 42.94 | 71.67 | 5 | 5 |
| CodeLlama-13B-Instr | 42.65 | 74.81 | 6 | 4 |
| StarCoder 2 7B | 33.60 | 65.74 | 7 | 7 |
| Phi-4 Reasoning (14B) | 18.14 | 46.30 | 8 | 8 |
| DeepSeek-R1-Distill-Qwen-7B | 17.22 | 44.63 | 9 | 9 |

Table C.3: Pass@1 (%) on HumanEval, MBPP, and the **Adj. accuracy** from SemBench.

| Family | Model Name | Type | Size (B) | HumanEval | MBPP | Adj. accuracy |
|---|---|---|---|---|---|---|
| GPT | GPT-4o Mini | General | – | 88.4 | – | 94.2 |
| | GPT-3.5 Turbo | General | – | 71.9 | – | 76.0 |
| DeepSeek | DeepSeek-Coder V2-Lite-Instr | Code | 16 | 81.1 | 68.8 | 70.9 |
| | DeepSeek-Coder 7B-Instr v1.5 | Code | 7 | 64.1 | 64.6 | 51.6 |
| | DeepSeek-R1-Distill-Qwen-7B | Reasoning | 7 | – | 17.2 | 10.3 |
| Llama | CodeLlama-13B-Instr | Code | 13 | 42.7 | 49.4 | 44.7 |
| | CodeLlama-7B-Instr | Code | 7 | 34.8 | 44.4 | 36.0 |
| | Llama-3 8B-Instr | General | 8 | 72.6 | – | 43.4 |
| Mistral | Mistral 7B-Instr (v0.3) | General | 7 | – | – | 67.0 |
| | Mamba Codestral 7B (v0.1) | Code | 7 | 75.0 | 68.5 | 52.7 |
| Qwen | Qwen2.5-Coder 14B-Instr | Code | 14 | 89.6 | 86.2 | 64.3 |
| | Qwen3 14B | General | 14 | – | 73.4 | 62.5 |
| StarCoder | StarCoder 2 7B | Code | 7 | 30.5 | 47.4 | 31.6 |
| Phi | Phi-4 Reasoning (14B) | Reasoning | 14 | – | 12.5 | 18.0 |

## C.4 Unexpected Results Analysis

Table C.4: Correlation between SemBench Adj. Accuracy and code generation benchmarks.

| Benchmark | $\rho$ | $p_\rho$ | $\tau$ | $p_\tau$ | $N$ |
|---|---|---|---|---|---|
| HumanEval | 0.770 | $9.22\times10^{-3}$ | 0.644 | $9.15\times10^{-3}$ | 10 |
| MBPP | 0.939 | $5.48\times10^{-5}$ | 0.822 | $3.58\times10^{-4}$ | 10 |

Almost all results in Section 4 followed expected trends that (i) code-specialised models outperform general-purpose ones and (ii) larger parameter counts generally improve scores. We highlight two notable exceptions: **Phi-4 Reasoning** and **CodeLlama-13B-Instr**.

**Phi-4 Reasoning — weak code scores are consistent with prior reports.** Microsoft's technical report shows that Phi-4 variants excel at maths and logic but lag on coding benchmarks (e.g., MBPP). Our SemBench findings (18.14 % accuracy) align with that pattern: the model was tuned with reasoning-centric synthetic data, not large volumes of real-world code, so its generation strategies transfer poorly to code tasks and code semantic questions.

**CodeLlama-13B-Instr vs. CodeLlama-7B-Instr — a modest, task-dependent gap.** Meta AI's release notes report only a 5 – 8 % advantage of the 13 B variant over the 7 B on HumanEval and MBPP. In SemBench, we observe a more nuanced picture: the 13B model

underperforms the 7B on Data Dependency questions (46.7% vs. 80.3% accuracy), though it outperforms or ties the 7B on other categories.

Table C.3 provides a possible explanation for the performance disorder between 7B and 13B models. When we refine SemBench to include only code generation-related categories, CodeLlama-13B-Instr achieves a better performance than CodeLlama-7B-Instr. This reveals that it is questions that are less correlated to code generation ability that play a role in the difference.

# D  ADDITIONAL RELATED WORK

## D.1  AUTOMATION DATASET CONSTRUCTION

Automation is key to scaling benchmarks for evaluating complex reasoning tasks like code semantics Han et al. (2024). Existing benchmarks, e.g., HumanEval, MBPP, APPS, depend heavily on manual annotation, limiting dataset size, diversity, and consistency. In contrast, fields like math and NLP have successfully used automation (e.g., GSM8K, MATH, SQuAD) to generate large, diverse datasets Cobbe et al. (2021); Hendrycks et al. (2021b); Clark et al. (2019); Rajpurkar et al. (2016). However, code understanding benchmarks lack such automation. Recent work like SeqCoBench explores semantics but still relies on human input. We address this gap by leveraging compiler infrastructure (ASTs and LLVM) to automatically generate diverse, accurate semantic questions and answers. This enables scalable, reproducible evaluation of LLMs' semantic reasoning, advancing the benchmarking of code understanding.

