# OpenReview forum: "Do LLMs Really Understand Code? A Semantic Benchmark with Automated Question Generation and Evaluation"
_ICLR.cc/2026/Conference — ICLR 2026 Conference Desk Rejected Submission_

### Official Review · Reviewer_EbGi · 2025-10-19

**Soundness:** 2
**Presentation:** 2
**Contribution:** 2
**Rating:** 2
**Confidence:** 5

**Summary:**

This paper presents SemBench, a benchmark designed to evaluate whether LLMs truly understand code semantics, through the lens of several tasks motivated by PL concepts. Models are asked with static analysis questions, which have ground truth answers by static analyzers.

The paper evaluates some popular LLMs and reveals the poor performance of these LLMs on such semantic understanding tasks. The results highlight that while LLMs are good at understanding local semantics such as control-flow tasks, they are bad at global ones like determining liveness. The paper also studies the performance correlation between semantic understanding and code generation, which finds that local understanding is likely to be included in existing benchmarks, but global analysis has not been evaluated yet.

**Strengths:**

- **Problem Formulation:** This paper tries to evaluate LLMs' performance of code semantic understanding at a finer-grained level. It probes the capabilities using tasks based on PL concepts, which can reveal new issues that are not covered by existing evaluation methods, like code generation. In general, this direction provides more insights into code LLMs.
- **Dataset Construction:** The paper constructs a dataset that is comprehensive to evaluate code semantics understanding, considering the number of samples and the variety of tasks.
- **Empirical Results:** The paper empirically presents some findings using the proposed benchmark, including the gap between local and global understanding, the correlation between semantics understanding and code generation, and the inconsistent capabilities across different tasks.

**Weaknesses:**

- **Lack of Strong SOTA LLMs:** This paper does not evaluate the performance of strong SOTA LLMs, such as GPT-5, Claude Sonnet 3.7 or 4, Gemini-2.5-Pro, etc. One may wonder if such strong LLMs already solve the semantic understanding problems in this paper pretty well. Current results are limited to small models.
- **Flaw of Prompts for Evaluation:** The CoT prompt demonstrated in Appendix B is WRONG. To leverage the power of CoT, one should ask the model to do reasoning BEFORE giving the final answer, otherwise the answer cannot benefit from the additional computational power of CoT. This flaw directly leads to the invalidation of almost all results in this paper... Because complex analysis does require more steps of reasoning before giving the answer, as indicated by [previous works](https://arxiv.org/abs/2310.07923) showing that CoT steps can fundamentally enhance the expressiveness of transformers. Therefore, the performance results obtained by no reasoning are not very meaningful, as they cannot reveal the true capability of LLMs. Also, without enough demonstration such as careful few-shot prompting, it is suspicious whether the description of each task can be well understood by the LLM, such as the definition of variable liveness.
- **Lack of Failure Modes Analysis:** Due to the flaw of the CoT prompt, there are also no valid reasoning steps that can be used to do in-depth analysis of why LLMs failed to do long-range program analysis.

**Questions:**

Please see the section of weaknesses above, which states the concerns.

---

### Official Review · Reviewer_6CsX · 2025-10-22

**Soundness:** 3
**Presentation:** 4
**Contribution:** 3
**Rating:** 6
**Confidence:** 3

**Summary:**

This paper investigates whether LLMs truly comprehend code semantics, despite their strong performance in code generation tasks, and how this impacts code quality. To probe this, the authors introduce SemBench, a novel benchmark comprising 1,000 diverse C programs sourced from the CodeParrot GitHub dataset, paired with 15,404 semantic questions across six core properties taught in undergraduate programming courses: function reachability, loop reachability, data dependency, variable liveness, dominator sets, and dead code. These properties, computable via deterministic algorithms (e.g., Clang AST and LLVM static analysis), enable precise, automated ground truth labeling.

**Strengths:**

- Pioneers a large-scale semantic benchmark for code understanding, filling a gap in existing evaluations that overlook direct semantic probing beyond syntax or pass rates; focuses on foundational, analyzable properties for reliable metrics.

- SemBench-QE leverages compiler tools (Clang/LLVM) for end-to-end automation, eliminating costly human labeling and handling LLM output uncertainties via structured prompts and parsers—ideal for low-resource languages like C.

- Broad evaluation across 14 diverse LLMs uncovers nuanced gaps (local vs. global semantics, self-generated code flaws) and practical correlations to generation benchmarks, guiding future LLM improvements like semantic-augmented training.

- Python extension validates generalizability; open-sourced dataset and pipeline foster community research in code reasoning.

**Weaknesses:**

- Primarily C-focused (Python extension is small at 540 questions), neglecting dynamic features in higher-level languages (e.g., Python introspection) or multi-file/modern paradigms (e.g., concurrency, optimizations); only six properties, omitting advanced semantics.

- Can this benchmark be extended to other programming languages? Would the finding be different if extended to other programming languages?

- Binary Q&A may undervalue multi-step reasoning (no multi-turn or advanced prompting); static analysis misses runtime behaviors (e.g., indirect calls via function pointers), potentially yielding conservative ground truths.

- The 1,000 programs generate 15k questions but remain modest vs. LLM-scale corpora; sourced from CodeParrot without quantified diversity (e.g., no KL-divergence on code styles), risking biases.

- Relies on pass@1 without ablations (e.g., prompt variants, few-shot), human baselines, or robustness tests; correlation analysis uses small N=10 with marginal p-values.

- LLM output parser may amplify noise in ambiguous responses; lacks discussion of computational costs, ethical concerns (e.g., dual-use in code vulnerabilities).

**Questions:**

Please refer to the weaknesses.

---

### Official Review · Reviewer_fjGX · 2025-11-01

**Soundness:** 2
**Presentation:** 2
**Contribution:** 3
**Rating:** 4
**Confidence:** 4

**Summary:**

This paper mainly presents a new benchmark SemBench, consisting 1000 diverse C programs with 15404 semantic questions spanning six fundamental properties. Besides, this paper evalaute 14 poplular LLMs across 7 families.

**Strengths:**

1. An interesting research direction with many cases.
2. Many interesting findings are presented, such as deficiency in understanding of current models.

**Weaknesses:**

1. Presentation. Overall, this template is not required as ICLR 2026. Authors should use the appropriate tempalte. Besides, the presentation and organisation of different sections in this paper are not good enough. Figures and tables are not precisely organized.
2. Detailed statistical analysis of this new benchmark is missing, which makes the reviewer very confused about this dataset.

**Questions:**

N/A

---

### Official Review · Reviewer_RXWa · 2025-11-03

**Soundness:** 3
**Presentation:** 2
**Contribution:** 1
**Rating:** 2
**Confidence:** 5

**Summary:**

This paper introduces SemBench, a benchmark designed to evaluate the semantic understanding of code by Large Language Models (LLMs). SemBench proposes a new evaluation methodology based on automated question generation from static program analysis, covering 6 static analysis tasks: function reachability, loop reachability, dominators, data dependency, variable liveness, and dead code.

**Strengths:**

## Soundness

**1. Comprehensive Experimental Design**

The experimental design is rigorous. The authors have evaluated a wide range of 14 popular LLMs, providing a comprehensive overview of the current state-of-the-art. The choice of six fundamental semantic properties is well-justified, as they are core concepts in program analysis and are taught in undergraduate compiler courses. The use of static analysis tools to generate the benchmark ensures a high degree of confidence in the ground truth, which is a critical aspect of any reliable evaluation.


## Effectiveness

**1. Clear Demonstration of LLM Deficiencies** The paper is highly effective in demonstrating the deficiencies of current LLMs in semantic understanding. The results are presented clearly and convincingly, with detailed breakdowns of performance across different models and semantic properties. The high error rates on many of the tasks, even for the most advanced models, provide a stark and compelling illustration of the limitations of current approaches. This will undoubtedly motivate further research into new architectures and training methods for improving the semantic reasoning capabilities of LLMs.

**Weaknesses:**

## Novelty

**1. Relation  to Existing Semantic Benchmarks**

The paper claims that SemBench is the first large-scale benchmark to evaluate LLMs’ semantic understanding of code. While SemBench is a valuable contribution, this claim is somewhat overstated. Other recent benchmarks also aim to evaluate semantic understanding, albeit with different methodologies. For instance, **EquiBench [1]** evaluates semantic understanding through program equivalence checking, a task that inherently requires deep semantic reasoning. Similarly, **CodeQueries [2]** uses queries from the static analysis tool CodeQL to create a dataset of semantic questions about code. Execution-based benchmarks like **CRUXEval (Gu et al., 2024) [6]** and **PLSemanticsBench (Thimmaiah et al., 2025) [7]** evaluate semantic understanding by testing a model's ability to reason about program execution and its outcome.


The paper presents SemBench as a novel way to evaluate LLMs, but the idea of using LLMs for static analysis tasks has been explored in prior work. For example, **Su & McMillan (2025) [3]** investigated the capability of LLMs for static analysis tasks like callgraph and dataflow generation, finding that LLMs perform poorly. Other works have also explored using LLMs for program analysis **(Li et al., 2023 [4]; Fang et al., 2024 [5])**. The paper would be stronger if it acknowledged this prior work and more clearly articulated how SemBench differs from and advances these previous efforts.


While SemBench’s focus on fundamental semantic properties and its automated question generation pipeline are novel, the paper would be stronger if it more accurately positioned itself within the existing landscape of semantic code understanding benchmarks. It would be better to frame SemBench as a novel and complementary approach to existing work, rather than claiming to be the first of its kind.

## Soundness

**1. Limited Scope of Analysis (Intra-procedural Only)**

The benchmark is limited to intra-procedural (function-level) analysis. This is a significant limitation, as real-world programs are composed of multiple functions and files that interact in complex ways. Understanding these interactions through inter-procedural and cross-file analysis is crucial for many software engineering tasks. By focusing only on function-level semantics, the benchmark may not accurately reflect the challenges of real-world code understanding. The authors should discuss the implications of this limited scope and the challenges of extending their approach to whole-program analysis.

**2. Potential Biases from Toolchain Dependence**

The benchmark generation process relies heavily on the Clang and LLVM toolchains for static analysis. While these are powerful and widely-used tools, they have their own limitations and idiosyncrasies. The choice of static analysis tools could introduce subtle biases into the benchmark. For example, the way Clang's AST is structured or how LLVM performs its analysis could influence the types of questions generated and the distribution of semantic properties in the benchmark. The paper would be more robust if it acknowledged this potential for toolchain-specific biases and perhaps discussed how the benchmark could be generalized or validated with other static analysis tools.

## Significance

**1. Unclear Justification for Static Analysis as a Measure of Semantic Understanding**

The paper does not provide a clear justification for why static analysis is a better measure of "semantic understanding" than execution-based approaches. These execution-based approaches like **CRUXEval (Gu et al., 2024) [6]** and **PLSemanticsBench (Thimmaiah et al., 2025) [7]** arguably provide a more direct measure of semantic understanding, as the semantics of a program are ultimately defined by its behavior during execution. The paper would be more convincing if it either provided a stronger argument for the primacy of static analysis in evaluating semantic understanding or positioned SemBench as a complementary approach to execution-based benchmarks.

**2. Basic Nature of Static Analysis Tasks**

The six semantic properties tested in SemBench are fundamental concepts in compiler theory, often taught in undergraduate courses. While it is concerning that LLMs struggle with these basic tasks, the limited complexity of these problems may not be representative of the challenges faced in real-world software development. The paper should discuss how performance on these basic tasks relates to more complex, real-world semantic understanding challenges.


## Effectiveness

**1. Limited Analysis of the Root Causes of LLM Failures**

The paper provides a thorough evaluation of LLM performance on SemBench, but the analysis of *why* the models fail is somewhat limited. The paper shows that LLMs struggle with certain semantic properties, but it does not delve deeply into the root causes of these failures. For example, are the failures due to a lack of architectural support for reasoning, insufficient or inappropriate training data, or an inability to understand the nuances of program semantics? It might be worthwhile to conduct a more fine-grained error analysis to categorize the types of mistakes the models are making. This could provide valuable insights for future research on improving the semantic reasoning capabilities of LLMs.



## References

[1] [EquiBench: Benchmarking Large Language Models’ Reasoning about Program Semantics via Equivalence Checking](https://arxiv.org/html/2502.12466)

[2] [CodeQueries: A Dataset of Semantic Queries over Code](https://dl.acm.org/doi/fullHtml/10.1145/3641399.3641408)

[3] [Do Code LLMs Do Static Analysis?](https://arxiv.org/abs/2505.12118)

[4] [The Hitchhiker's Guide to Program Analysis: A Journey with Large Language Models](https://arxiv.org/abs/2308.00245)

[5] [Large Language Models for Code Analysis: Do LLMs...](https://www.usenix.org/conference/usenixsecurity24/presentation/fang)

[6] [CRUXEval: A Benchmark for Code Reasoning, Understanding and Execution](https://arxiv.org/abs/2401.03065)

[7] [PLSemanticsBench: Large Language Models As Programming Language Interpreters](https://arxiv.org/abs/2510.03415)

**Questions:**

## Clarification Questions

1.  In Section 4.2, the paper states that "the overall accuracy on self-produced code does not improve over SemBench." Could the authors elaborate on this finding? What's the difference between self-produced code and others? A more detailed analysis of this result would be beneficial.

## Discussion Questions

1.  What are the authors' thoughts on potential avenues for future research to address the deficiencies identified in this paper? For example, do they believe that new model architectures, pre-training objectives, or fine-tuning strategies are needed?


2.  Given that real-world code understanding often requires inter-procedural and cross-file analysis, how do the authors envision SemBench evolving to address these more complex scenarios? What are the main challenges in extending the current methodology to a whole-program analysis setting?

3.  The paper argues that static analysis is a good proxy for semantic understanding. Could the authors elaborate on why they believe this is a more effective approach than execution-based benchmarks like CRUXEval or PLSemanticsBench, which directly test a model's ability to reason about program behavior? What are the trade-offs between these two evaluation paradigms?

---

### Note · Program_Chairs · 2026-01-17
**Submission Desk Rejected by Program Chairs**

The following references in this submission do not refer to real documents and/or have major errors in bibliographic information:

 Yuchen Zan, Rui Wang, Mingjun Shen, and Qian Li. 2025. Benchmark Correlation Analysis for Large Language Models. Proceedings of the AAAI Conference on Artificial Intelligence (2025).